# QuantiShift: Any-Shift Object Quantification by Text-to-Image Diffusion Models

## Abstract

Accurately quantifying objects with text-to-image diffusion models remains challenging, especially under distribution shifts. Existing methods struggle to maintain numerical precision across varying object categories, count distributions, and visual domains. To address this, we propose *QuantiShift*, a shift-aware prompting framework that adapts to different shift types without diffusion model retraining. *QuantiShift* introduces shift-aware prompt optimization, where distinct prompt components explicitly tackle number shifts, label shifts, and covariate shifts, ensuring precise object quantification across varying distributions. To further enhance generalization, we propose consistency-guided any-shift prompting, which enforces alignment between textual prompts and generated images by mitigating inconsistencies caused by distribution shifts. Finally, we develop hierarchical prompt optimization, a two-stage refinement process that first adapts prompts to individual shifts and then calibrates them for cross-shift generalization. To evaluate robustness, we introduce a new benchmark designed to assess object quantification under diverse shifts. Extensive experiments demonstrate that *QuantiShift* achieves state-of-the-art performance, considerably improving accuracy and robustness over existing methods.

## 1 Introduction

Text-to-image generative models have made impressive strides in producing high-quality images from textual descriptions, *e.g.*, (Paiss et al., 2023; Starr et al., 2013; Dao et al., 2023; Yang et al., 2024; Chen et al., 2024). However, precisely quantifying objects within these generated images remains a challenging task (Zafar et al., 2024; Binyamin et al., 2025; Kang et al., 2025). Zafar et al. (2024) posed the problem of object quantification and introduced a prompt learning approach to address the problem, laying the foundation for subsequent works. Building on this idea, Sun et al. (2024) quantifies objects under domain shifts. However, their QUOTA framework primarily focuses on domain generalization and does not explicitly handle other types of distribution shifts, such as number shifts and label shifts. These limitations highlight a critical gap in achieving robust, shift-aware object quantification, particularly for generative models. In this paper, we propose an object quantification method that is designed to be robust to any shift in prompt distributions.

Any-shift prompting has been introduced by Xiao et al. (2024) as a probabilistic inference framework to improve generalization across multiple distribution shifts in vision-language models. By explicitly modeling the relationship between training and test distributions, it constructs hierarchical training and test prompts to enhance adaptation. However, existing any-shift prompting methods are primarily designed for classification tasks and have not been explored for object quantification in text-to-image generation. Inspired by their work, we introduce *QuantiShift*, the first any-shift object quantification framework for text-to-image diffusion models. Unlike prior work, *QuantiShift* explicitly models shift-aware prompting in the context of generative models, addressing number shifts, label shifts, and covariate shifts. Any-shift prompting focuses on classification tasks, modeling distribution shifts through hierarchical prompt structures. In contrast, *QuantiShift* directly learns shift-aware prompt embeddings for generative models, enabling precise object quantification without requiring test-time adaptation.

In this work, we make three key contributions. *First*, we introduce a shift-aware prompt optimization framework that explicitly models number, label, and covariate shifts using dedicated prompt tokens.

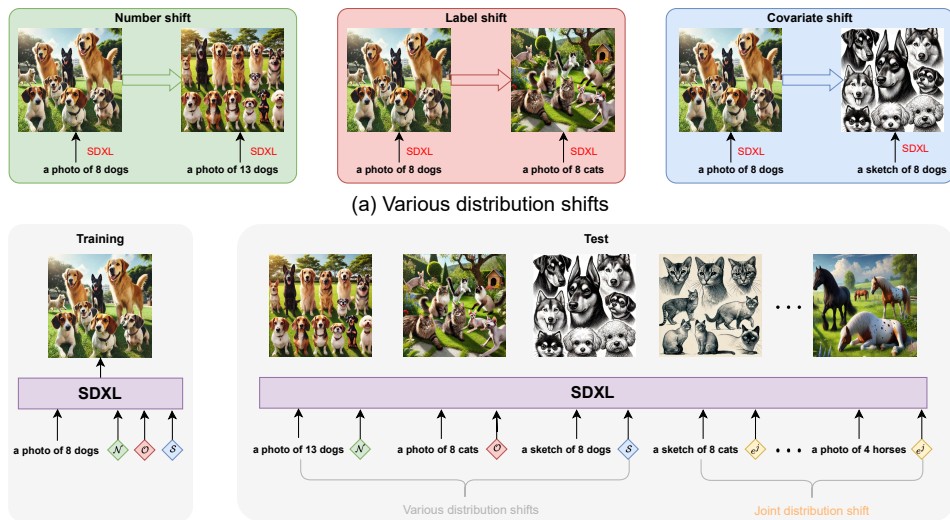

(a) Various distribution shifts

(b) Generalization over distributions by any-shift prompting

Figure 1: **Distribution shifts in text-to-image diffusion models.** (a) Examples of different shift types, including number shifts, label shifts, and covariate shifts. (b) Our proposed *QuantiShift* leverages shift-aware prompting to jointly address individual distribution shifts and their combinations, considerably enhancing object quantification robustness.

This structured design disentangles different shift factors, enabling targeted adaptation without retraining. *Second*, we propose consistency-guided prompting to enhance robustness by enforcing alignment between textual prompts and generated images across varying linguistic and visual conditions, thereby mitigating discrepancies between intended object counts and generated outputs. *Third*, we present hierarchical prompt optimization (HPO), a two-stage refinement strategy that first adapts prompts to individual shifts and then calibrates them for cross-shift generalization. By jointly optimizing prompt representations at both the local and global levels, HPO consistently improves numerical accuracy and robustness under unseen distribution shifts. Figure 1 illustrates these distribution shifts and their impact on text-to-image generation, where conventional models struggle to maintain numerical accuracy under varying conditions. Our proposed *QuantiShift* leverages shift-aware prompting to jointly address individual shifts and their combinations, significantly enhancing robustness in object quantification across domains. We also introduce a new benchmark specifically designed to evaluate object quantification under number shifts, label shifts, and covariate shifts, enabling a rigorous and standardized analysis of robustness and generalization in text-to-image diffusion models.

## 2 RELATED WORK

**Text-to-image generation.** Image generation has advanced significantly, evolving from early GAN-based methods (Goodfellow et al., 2014; Li et al., 2019b; Qiao et al., 2019a;b; Tao et al., 2022; Li et al., 2019a; Zhang et al., 2018) to diffusion-based models (Ho et al., 2020; Crowson et al., 2022; Ramesh et al., 2021; Gafni et al., 2022; Jain et al., 2022). Recent text-to-image diffusion models (Ramesh et al., 2022; Saharia et al., 2022; Höllein et al., 2024; Qu et al., 2024; Jiang et al., 2024; Ding et al., 2022) have demonstrated impressive image-generation capabilities, enabling high-quality synthesis from textual prompts. Flow matching models have emerged as an alternative approach to generative modeling, offering advantages in training stability and sample quality (Lipman et al., 2023; Hu et al., 2024; Chen & Lipman, 2023). While these methods have shown promise in controllable generation tasks, precise object quantification remains a challenge. Similar to diffusion models, they struggle to maintain numerical accuracy, particularly under distribution shifts. Our work addresses this gap by introducing shift-aware prompting techniques to improve robustness in text-to-image diffusion models.

**Controllable and personalized image generation.** Controlling image attributes during generation is a fundamental challenge. Textual Inversion (Gal et al., 2023) and DreamBooth (Ruiz et al., 2023) enable personalization by fine-tuning models on new concepts. However, while methods like Textual Inversion and DreamBooth focus on qualitative personalization, other approaches (Pang et al., 2024;

Lin et al., 2024) further explore controllable generation but still lack explicit numerical control. Existing numerical control approaches (Kajić et al., 2024; Yi et al., 2024; Sohn et al., 2023) often rely on prompt engineering, which may be sensitive to distribution shifts, potentially limiting robustness in real-world scenarios. Our work introduces shift-aware prompt optimization to achieve precise object quantification without retraining, extending personalized generation techniques to numerical control across different visual domains.

**Prompt optimization for image consistency.** Ensuring consistency between text prompts and generated images has been explored through various prompt optimization techniques. Methods like Prompt-to-Prompt (Hertz et al., 2023) and Null-inversion (Mokady et al., 2023) manipulate attention mechanisms to refine generated content, while Attend-and-Excite (Chefer et al., 2023) ensures all mentioned objects appear in the output. However, these approaches were not designed for numerical control and do not explicitly enforce object count accuracy in generated images. Our work introduces a structured, shift-aware prompting framework with hierarchical prompt optimization that enables precise object quantification across diverse prompts and visual styles, substantially improving numerical consistency in generated images.

## 3 PROBLEM STATEMENT

Accurately quantifying objects in text-to-image diffusion models remains a fundamental challenge, particularly under distribution shifts where object categories, counts, or visual styles differ from those seen during training. These shifts often cause models to generate incorrect object counts, limiting their reliability for applications requiring numerical precision. To systematically address this issue, we define three key types of distribution shifts that impact object quantification.

**Distribution shifts in object quantification.** We formalize object quantification shifts using three primary factors:

- **Number shift** ($\mathcal{N}$): Variation in numerical counts of objects, requiring models to generalize to unseen quantities.
- **Label shift** ($\mathcal{O}$): Differences in object categories, testing the model's ability to quantify novel objects.
- **Covariate shift** ($\mathcal{S}$): Changes in visual styles, assessing robustness to stylistic variations.

**Training and evaluation setup.** To evaluate model generalization under these shifts, we divide the dataset into disjoint training and evaluation sets:

$$\mathcal{D}^{\text{base}} = (\mathcal{S}^{\text{base}}, \mathcal{N}^{\text{base}}, \mathcal{O}^{\text{base}}) \tag{1}$$

$$\mathcal{D}^{\text{new}} = (\mathcal{S}^{\text{new}}, \mathcal{N}^{\text{new}}, \mathcal{O}^{\text{new}}) \tag{2}$$

where $\mathcal{D}^{\text{base}}$ is used for training, and $\mathcal{D}^{\text{new}}$ is reserved for evaluation. We enforce strict separation:

$$\mathcal{S}^{\text{base}} \cap \mathcal{S}^{\text{new}} = \emptyset, \mathcal{N}^{\text{base}} \cap \mathcal{N}^{\text{new}} = \emptyset, \mathcal{O}^{\text{base}} \cap \mathcal{O}^{\text{new}} = \emptyset. \tag{3}$$

This ensures that models are tested on entirely new conditions without relying on memorization.

**Challenges in text-to-image quantification.** Unlike classification tasks (Zhou et al., 2022a;b), where models predict discrete labels, object quantification requires precise numerical consistency between textual prompts and generated images. However, diffusion models often misinterpret numerical specifications, leading to incorrect object counts, particularly under distribution shifts. This highlights the need for explicit shift-aware mechanisms to enhance numerical precision and robustness.

## 4 METHODS

Text-to-image diffusion models often miscount under distribution shifts in number, category, and style. We address this with *QuantiShift*, a prompting framework that improves numerical fidelity and transfer *without retraining the diffusion model*. The framework comprises three complementary pieces. First, *shift-aware prompt optimization* factorizes control into disentangled tokens for number, class, and style, enabling targeted handling of number/label/covariate shifts (§4.1). Second, *consistency-guided*

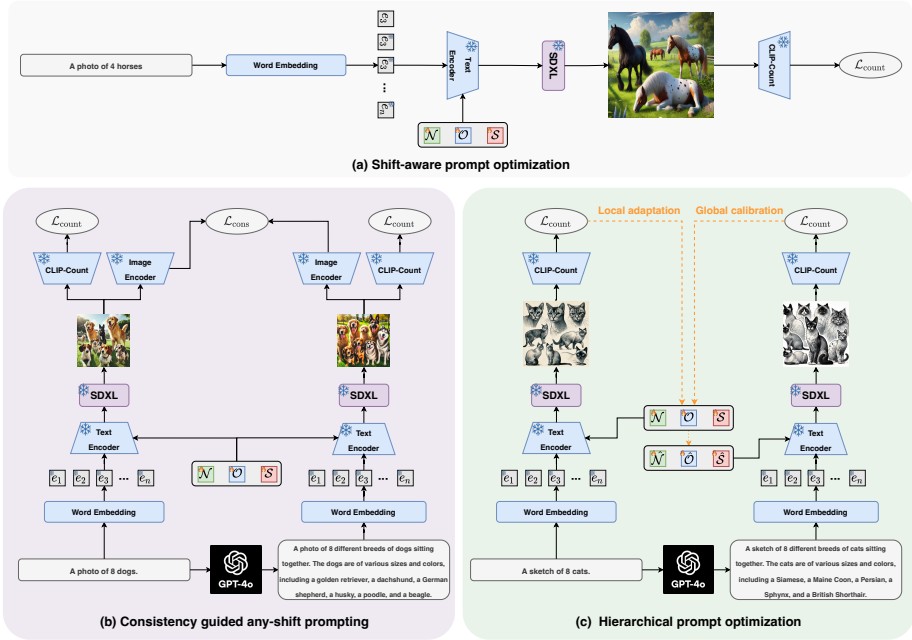

**Figure 2:** *QuantiShift* **prompting framework for object quantification under distribution shifts.** (a) *Shift-aware prompt optimization*: dedicated tokens for number ($\mathcal{N}$), class ($\mathcal{O}$), and style ($\mathcal{S}$) explicitly encode shift factors, enabling structured adaptation in diffusion models. (b) *Consistency-guided prompting*: alignment between textual descriptions and generated images is enforced (via a frozen CLIP-Count estimator), improving robustness across domains. (c) *Hierarchical Prompt Optimization (HPO)*: a two-stage refinement that first adapts prompts to each observed shift (local stage) and then calibrates them for cross-shift generalization (global stage), without retraining the diffusion model.

*prompting* regularizes prompts by enforcing paraphrase- and style-invariant text–image alignment, stabilizing counts across wording and domains (§4.2). Third, *Hierarchical Prompt Optimization (HPO)* casts prompt refinement as a two-stage objective—local adaptation followed by global calibration—to strengthen cross-shift generalization (§4.3). We detail each component below.

## 4.1 SHIFT-AWARE PROMPT OPTIMIZATION

Our shift-aware prompt optimization approach constructs prompts using structured representations of object count, category, and style, ensuring adaptability to unseen distributions.

**Prompt representation.** We formulate a prompt as a structured sequence: $P = \langle A, S, \text{of } N \text{ CLASS}, O \rangle$, where $\langle A \rangle$ is a fixed template, $\langle S \rangle \in \mathcal{S}$ denotes the style, $\langle N \rangle \in \mathcal{N}$ specifies the object count, and $\langle O \rangle \in \mathcal{O}$ represents the object category. To ensure adaptability across diverse shifts, we introduce a shift-aware token space $\mathcal{T}$, which encapsulates three learnable shift-specific tokens.

**Shift-specific learnable tokens.** To systematically handle different distribution shifts, we define $\mathcal{T}$ as: $\mathcal{T} = \{\mathcal{N}, \mathcal{O}, \mathcal{S}\}$ where:

- Number token ($\mathcal{N}$): Captures numerical variations to mitigate *number shift*.
- Class token ($\mathcal{O}$): Encodes object category-specific features to address *label shift*.
- Style token ($\mathcal{S}$): Embeds domain-specific characteristics to adapt to *covariate shift*.

While our framework focuses on these three primary shift types, it can be extended to incorporate additional shifts, such as positional or compositional variations, by introducing corresponding learnable tokens. This structured token representation allows for explicit modeling of distribution shifts, improving object quantification robustness in text-to-image diffusion models.

During training, we optimize the prompt embeddings to minimize object quantification error across seen shifts while ensuring generalization to unseen conditions. Specifically, we sample prompts from

the training set $\mathcal{D}^{\text{base}}$: $P_i = \langle A, S_i^{\text{base}}, \text{of } N_i^{\text{base}} \ \texttt{CLASS}, O_i^{\text{base}} \rangle$ and use a pre-trained CLIP-Count model (Jiang et al., 2023) to obtain object count estimates. The objective function for prompt optimization is defined as:

$$\mathcal{L}_{\text{count}} = \sum_i \| f_{\text{cnt}}(G(P_i)) - N_i \|^2, \tag{4}$$

where $G(\cdot)$ represents the text-to-image diffusion model and $f_{\text{cnt}}(\cdot)$ is the counting function. By minimizing $\mathcal{L}_{\text{count}}$, we refine prompts to improve object quantification accuracy. As illustrated in Figure 2 (a), this process is part of our shift-aware prompt optimization framework, where distinct prompt tokens explicitly model number shifts, label shifts, and covariate shifts.

## 4.2 Consistency-guided any-shift prompting

While shift-aware prompt optimization enables the model to handle individual distribution shifts, ensuring robustness across unseen conditions remains a challenge. Standard prompting methods in text-to-image diffusion models often struggle to maintain numerical accuracy and semantic consistency under distribution shifts. To address this, we introduce consistency-guided any-shift prompting, a mechanism that reinforces alignment between textual prompts and generated images, improving generalization across shifts. By enforcing consistency constraints, our approach enhances robustness to prompt rewording and shift perturbations, ensuring more stable object quantification.

**Consistency regularization.** To enforce consistency across different prompt formulations, we introduce an auxiliary textual variation generated by GPT-4o (OpenAI, 2023): $P_i^{\text{alt}} = \text{Trans}(P_i)$, where $\text{Trans}(\cdot)$ represents a transformation function that paraphrases the original prompt $P_i$ while preserving its semantic meaning. Furthermore, $P_i^{\text{alt}}$ incorporates the same learnable shift-aware tokens as the original prompt, including the number, class, and covariate tokens, as shown in Figure 2. This ensures that both prompt variants share identical shift-specific representations, reinforcing consistency at both the linguistic and embedding levels.

Given the original prompt $P_i$ and its reworded counterpart $P_i^{\text{alt}}$, we generate corresponding images using the text-to-image diffusion model:

$$I_i = G(P_i), \quad I_i^{\text{alt}} = G(P_i^{\text{alt}}). \tag{5}$$

To ensure alignment between $I_i$ and $I_i^{\text{alt}}$, we define a contrastive consistency loss:

$$\mathcal{L}_{\text{con}} = 1 - \cos(I_i, I_i^{\text{alt}}), \tag{6}$$

where $\cos(\cdot, \cdot)$ represents the cosine similarity between image embeddings extracted from a frozen image encoder. This loss penalizes unintended variations between images generated from semantically equivalent prompts, reinforcing prompt consistency under distribution shifts.

Consistency-guided any-shift prompting is integrated into our shift-aware prompt optimization framework. The total training objective combines the count loss from object quantification and the consistency loss to ensure both numerical precision and shift robustness:

$$\mathcal{L} = \mathcal{L}_{\text{count}} + \lambda_{\text{con}} \mathcal{L}_{\text{con}}, \tag{7}$$

where $\lambda_{\text{con}}$ is a weighting coefficient balancing the two objectives. Our consistency-guided any-shift prompting, shown in Figure 2 (b), enforces alignment between prompts and generated images, enhancing numerical accuracy and improving generalization to unseen distribution shifts in text-to-image diffusion models.

## 4.3 Hierarchical Prompt Optimization

To further improve generalization under unseen distribution shifts, we introduce *Hierarchical Prompt Optimization (HPO)*, a two-stage (bi-level) objective that refines shift-aware prompt embeddings across diverse conditions *without* retraining the text-to-image diffusion model. Conventional prompt learning optimizes prompts for a fixed training distribution, which limits transfer. HPO explicitly separates (i) *local, shift-specific adaptation* from (ii) *global, cross-shift calibration*, yielding prompts that remain numerically faithful beyond the training shifts.

**Two-stage objective.** HPO is formulated as a bi-level optimization with an inner objective for local adaptation and an outer objective for cross-shift generalization.

**Local adaptation.** Given a training distribution $\mathcal{D}^{\text{base}}$, we adapt the shift-aware tokens to each observed condition: $P_i = \langle A, S_i^{\text{base}}, \text{of } N_i^{\text{base}} \text{ CLASS}, O_i^{\text{base}} \rangle$, and minimize the counting loss

$$\mathcal{L}_{\text{local}} = \sum_i \left\| f_{\text{cnt}}(G(P_i)) - N_i \right\|^2. \tag{8}$$

**Global calibration.** To promote robustness under prompt rewording and unseen shifts, we evaluate the inner-updated tokens on an auxiliary textual variation $P_i^{\text{alt}}$ and minimize

$$\mathcal{L}_{\text{global}} = \sum_i \left\| f_{\text{cnt}}(G(P_i^{\text{alt}})) - N_i \right\|^2. \tag{9}$$

**Bi-level optimization.** Let $\mathcal{T}$ denote all learnable prompt embeddings (number/class/style tokens). HPO solves

$$\min_{\mathcal{T}} \quad \mathcal{L}_{\text{global}}(\mathcal{T}') \quad \text{s.t.} \quad \mathcal{T}' = \mathcal{T} - \alpha \nabla_{\mathcal{T}} \mathcal{L}_{\text{local}}(\mathcal{T}), \tag{10}$$

where $\alpha$ is the inner update step size. This hierarchical objective encourages inner-stage improvements that translate into stronger outer-stage generalization across shifts.

As illustrated in Fig. 2(c), HPO first adapts the shift-aware tokens to the observed condition (local stage) and then calibrates them for cross-shift generalization (global stage), leading to more accurate and robust object quantification without modifying the diffusion or text encoders.

## 5 EXPERIMENTS

### 5.1 EXPERIMENTAL SETUP

**Benchmark.** We introduce *QSBench*, a new benchmark specifically designed to evaluate the robustness of text-to-image diffusion models under number shift, label shift, and covariate shift. Unlike existing datasets that primarily assess image quality and text-image alignment, *QSBench* focuses on measuring how well a model adheres to numerical, categorical, and stylistic constraints specified in the prompt. *QSBench* builds upon QUANT-bench (Sun et al., 2024) but extends it beyond covariate shifts to comprehensively evaluate object quantification across multiple distribution shifts. While QUANT-bench primarily addresses domain generalization and is limited to 19 categories due to its reliance on YOLO-based object detection, *QSBench* introduces a broader evaluation setting with 147 object categories and explicitly models number shifts, label shifts, and covariate shifts. *QSBench* is derived from FSC-147 (Ranjan et al., 2021) and systematically evaluates object quantification across disjoint training and evaluation splits. The base subset consists of 74 object categories and 12 numerical values, while the new subset contains 73 distinct categories and 13 numerical values, ensuring no overlap between training and evaluation. For covariate shift assessment, we adopt a leave-one-out setting, where the model is trained on three visual styles (*Photo, Painting, Cartoon, Sketch*) and evaluated on the remaining unseen style. This structured evaluation allows for a rigorous assessment of model robustness across diverse distribution shifts.

**Metrics.** To evaluate quantification accuracy, we use Mean Absolute Error (MAE) and Root Mean Square Error (RMSE) based on an evaluation variant of CLIP-Count (Jiang et al., 2023), which estimates object counts from multiple localized patches. MAE measures the average absolute deviation from the target count, while RMSE penalizes larger errors through squared deviations. Since the model is trained on the Base subset and evaluated on both Base and New subsets, we compute the harmonic mean (H) of MAE and RMSE across Base and New to comprehensively assess both performance and generalization.

**Implementation details.** We conducted training and evaluation on a single NVIDIA L20 GPU with 48GB of memory. Each experiment consists of 5 epochs, with 2,664 iterations per epoch, totaling 13,320 iterations per experiment. Training each iteration takes approximately 0.1 minutes, leading to a total training time of around 22.2 hours. For image quality, we find a single denoising step is sufficient. The optimized quantification token can be reused without additional optimization time. We set the learning rate at 0.01 for optimization, and the CLIP-Count (Jiang et al., 2023) scaling hyperparameter $\lambda_{\text{scale}}$ to 0.6 for a static scale. Additional details on datasets and experimental settings can be found in the supplementary material. We will make our code available.

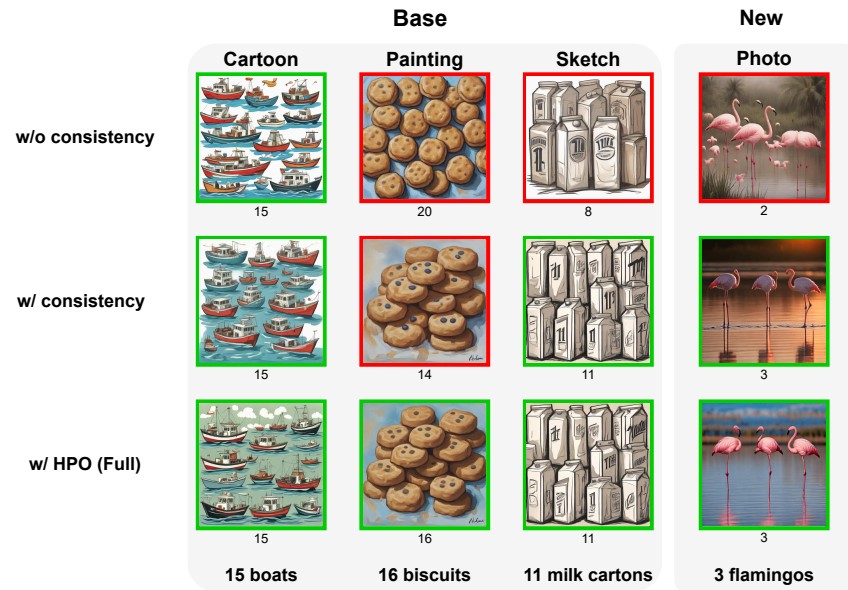

**Figure 3: Effect of consistency-guided prompting and Hierarchical Prompt Optimization (HPO).** Enforcing text–image consistency improves numerical accuracy across styles but can still fail on harder cases (red frames indicate incorrect counts; green frames indicate correct counts). Adding HPO, a two-stage refinement that first adapts locally and then calibrates globally, further strengthens cross-shift generalization and yields more precise counts on both *Base* and *New* domains.

## 5.2 RESULTS

**Effect of shift-aware tokens.** To analyze the contribution of each shift-aware token, we conduct an ablation study by selectively removing each token while keeping the other components unchanged. Table 1 reports the results on Base, New, and Harmonic Mean (H) subsets using MAE and RMSE as evaluation metrics, while Figure 7 (See Appendix) provides a qualitative comparison of generated images under different token configurations. The results indicate that incorporating any individual shift-aware token improves performance over the baseline SDXL model. Specifically, the number token helps control object count variations, the class token aids in category-specific adaptations, and the covariate token enhances robustness to style variations. However, relying on a single token leads to suboptimal performance under certain distribution shifts. As shown in Figure 7, using only a single token often results in inaccurate object counts, while the combination of all tokens ensures better numerical precision and consistency across different domains. The full model, incorporating all three tokens, achieves the lowest error across all settings, demonstrating the importance of jointly optimizing shift-aware tokens for effective generalization. These findings validate our shift-aware prompt optimization strategy, showing that a structured representation of shift factors significantly enhances object quantification accuracy in text-to-image diffusion models.

| $\mathcal{N}$ | $\mathcal{O}$ | $\mathcal{S}$ | Base | | New | | H | |
|---|---|---|---|---|---|---|---|---|
| | | | MAE ↓ | RMSE ↓ | MAE ↓ | RMSE ↓ | MAE ↓ | RMSE ↓ |
| | | | 17.83 | 31.99 | 17.52 | 43.80 | 17.32 | 36.07 |
| ✓ | | | 12.79 | 21.95 | 15.58 | 42.29 | 13.76 | 28.06 |
| ✓ | ✓ | | 15.23 | 27.92 | 16.53 | 44.63 | 15.75 | 34.09 |
| ✓ | | ✓ | 10.59 | **18.90** | 17.90 | 54.79 | 13.18 | 27.86 |
| ✓ | ✓ | ✓ | **10.27** | 19.03 | **13.95** | **35.93** | **12.53** | **24.55** |

**Table 1: Impact of different shift-aware tokens.**

**Effect of consistency-guided any-shift prompting.** We analyze the impact of consistency regularization by comparing models with and without the consistency constraint in Table 2. Enforcing consistency improves MAE and RMSE across all subsets, suggesting enhanced stability in object quantification. The results indicate that consistency constraints help maintain numerical precision under shift conditions, reducing discrepancies between intended and generated outputs, as illustrated in Figure 3. Figure 3 further illustrates the benefits of consistency-guided any-shift prompting. The left column shows images generated using simple prompts, where object counts are often inaccurate due to ambiguity in prompt interpretation. In contrast,

| Methods | Base | | New | | H | |
|---|---|---|---|---|---|---|
| | MAE ↓ | RMSE ↓ | MAE ↓ | RMSE ↓ | MAE ↓ | RMSE ↓ |
| Shift-aware tokens | 11.38 | 19.03 | 13.95 | 35.93 | 12.53 | 24.55 |
| w/ consistency | 10.58 | 15.89 | 14.13 | 35.86 | 12.05 | 21.81 |
| w/ HPO (full) | **7.30** | **9.36** | **7.29** | **9.03** | **7.26** | **9.12** |

**Table 2: Ablation of consistency-guided prompting and hierarchical prompt optimization.**

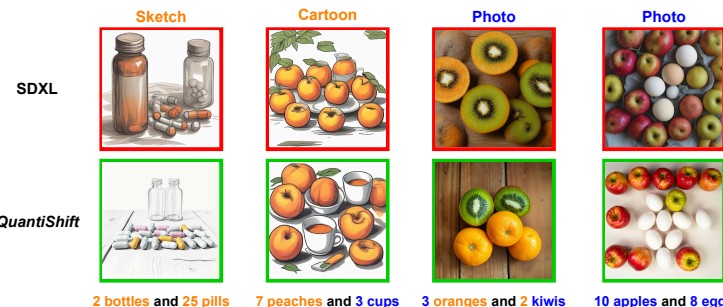

Sketch    Cartoon    Photo    Photo

SDXL

QuantiShift

**2 bottles** and **25 pills**    **7 peaches** and **3 cups**    **3 oranges** and **2 kiwis**    **10 apples** and **8 eggs**

**Figure 4: Multi-class object quantification results.** Orange labels indicate that the domain, number, and object class originate from the Base set, while blue labels indicate they come from the New set. *QuantiShift* accurately quantifies multiple object categories while ensuring numerical precision, outperforming SDXL (Podell et al., 2024) in handling diverse object compositions across different domains.

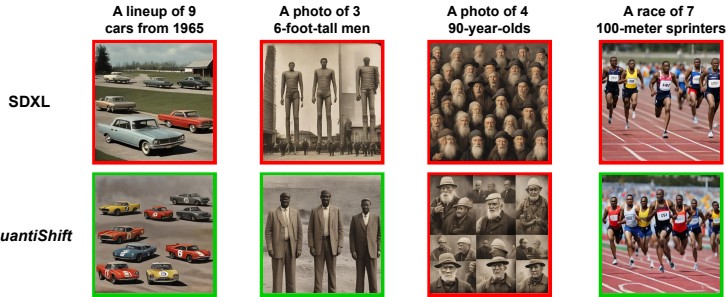

A lineup of 9 cars from 1965    A photo of 3 6-foot-tall men    A photo of 4 90-year-olds    A race of 7 100-meter sprinters

SDXL

QuantiShift

**Figure 5: Effect of number token on explicit and implicit numerical attributes.** Our method maintains accurate explicit counts while preserving implicit numerical attributes, demonstrating that the learned number token does not interfere with implicit numbers.

| Metrics | Methods | Photo | | | Painting | | | Cartoon | | | Sketch | | | Average | | |
|---|---|---|---|---|---|---|---|---|---|---|---|---|---|---|---|---|
| | | Base | New | H | Base | New | H | Base | New | H | Base | New | H | Base | New | H |
| MAE↓ | SDXL (Podell et al., 2024) | 15.77 | 19.29 | 17.35 | 18.32 | 17.35 | 17.82 | 17.52 | 21.71 | 19.39 | 19.69 | 11.74 | 14.71 | 17.82 | 17.52 | 17.32 |
| | IoCo (Zafar et al., 2024) | 11.98 | 16.23 | 13.79 | 13.13 | 15.24 | 14.11 | 11.56 | 19.72 | 14.57 | 14.48 | 11.11 | 12.57 | 12.79 | 15.58 | 13.76 |
| | QUOTA (Sun et al., 2024) | 10.21 | 14.25 | 11.90 | 11.93 | 12.86 | 12.38 | 9.75 | 14.21 | 11.56 | 10.73 | 9.21 | 9.91 | 10.66 | 12.63 | 11.44 |
| | *QuantiShift* | **7.72** | **8.79** | **8.22** | **6.99** | **7.10** | **7.04** | **6.75** | **7.02** | **6.88** | **7.71** | **6.24** | **6.89** | **7.30** | **7.29** | **7.26** |
| RMSE↓ | SDXL (Podell et al., 2024) | 27.00 | 39.52 | 32.08 | 33.24 | 47.86 | 39.23 | 32.56 | 60.73 | 42.40 | 35.14 | 27.08 | 30.59 | 31.99 | 43.80 | 36.07 |
| | IoCo (Zafar et al., 2024) | 21.13 | 35.23 | 26.41 | 21.92 | 44.77 | 29.43 | 20.36 | 61.88 | 30.64 | 24.40 | 27.28 | 25.76 | 21.95 | 42.29 | 28.06 |
| | QUOTA (Sun et al., 2024) | 18.54 | 23.71 | 20.81 | 18.25 | 27.56 | 21.96 | 16.32 | 40.32 | 23.24 | 20.21 | 21.25 | 20.72 | 18.33 | 28.21 | 21.68 |
| | *QuantiShift* | **9.79** | **10.63** | **10.19** | **11.36** | **8.78** | **9.39** | **8.45** | **8.64** | **8.55** | **10.41** | **7.43** | **8.67** | **9.36** | **9.03** | **9.12** |

**Table 3: Comparison with state-of-the-art for object quantification across *QuantShift* four visual styles: Photo, Painting, Cartoon, and Sketch.** *QuantiShift* achieves the lowest errors across all domains, demonstrating superior robustness and generalization.

the right column demonstrates the effect of consistency-enforced descriptions, which lead to more precise object quantification. The structured descriptions guide the model to generate images with the correct number of objects by reinforcing alignment between textual cues and visual representations. This validates that enforcing consistency during prompting enhances robustness, ensuring more reliable numerical adherence across different styles and object categories.

**Effect of hierarchical prompt optimization.** To further improve generalization, we incorporate hierarchical prompt optimization (HPO), which refines the shift-aware tokens across diverse conditions. Table 2 shows that HPO substantially reduces both MAE and RMSE compared to the consistency-only variant, with gains consistent on *Base*, *New*, and their harmonic mean. Figure 3 illustrates this trend: without consistency, the model often miscounts in the *New* domain (e.g., generating "2 flamingos" instead of "3"); adding consistency improves accuracy but still fails on harder cases. With HPO, local adaptation followed by global calibration yields the most reliable counts, enhancing numerical precision and robustness under diverse shifts.

**Multi-class object quantification analysis.** Figure 4 illustrates the effectiveness of *QuantiShift* in accurately quantifying multiple object categories across different domains, compared to the SDXL baseline. The results highlight two key observations. First, *QuantiShift* successfully maintains numerical precision when generating multi-class objects, whereas SDXL often produces incorrect object counts, as indicated by the red bounding boxes. Second, *QuantiShift* demonstrates the

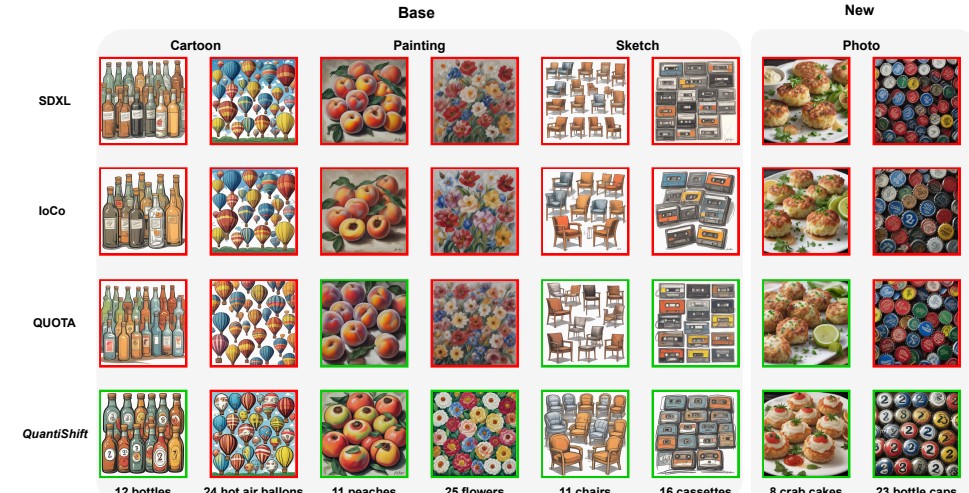

**Figure 6: Qualitative comparison of object quantification performance across different visual styles.** The red bounding boxes highlight failure cases where models generate incorrect object counts. *QuantiShift* produces more accurate object quantities across diverse domains, demonstrating improved generalization to unseen shifts. generalization across both base and new distributions, as evidenced by its ability to correctly generate object compositions with varying domain, number, and class configurations. These results validate the robustness of our shift-aware prompt optimization strategy in handling complex multi-object generation scenarios.

**Handling explicit and implicit numerical attributes.** Figure 5 evaluates whether the learned number token correctly handles explicit numerical values without interfering with implicit numerical attributes. Each prompt contains two numbers: one explicitly defining the quantity of objects (e.g., "4 people", "9 cars") and another implicitly describing an attribute (e.g., "90-year-olds", "1965 cars"). While *QuantiShift* effectively preserves both explicit and implicit numerical references in most cases, some challenges remain. For example, in the third case, where the prompt specifies "A photo of 4 90-year-olds", the model generates a larger group of elderly individuals instead of the exact count. This limitation arises from the entanglement between explicit count constraints and implicit demographic attributes in large-scale vision-language models. Future work will explore refining prompt embeddings to better disentangle numerical constraints from contextual attributes, ensuring more precise object quantification.

**Comparison with state-of-the-art.** We compare our proposed *QuantiShift* with three baseline models, SDXL (Podell et al., 2024), IoCo (Zafar et al., 2024) and QUOTA (Sun et al., 2024), on object quantification across different visual styles in *QSBench* (Table 3). Our method consistently achieves the lowest errors across all domains, significantly outperforming prior works. These results highlight the effectiveness of our shift-aware prompt optimization and hierarchical prompt optimization, which enable more precise and generalizable object quantification in text-to-image diffusion models. Figure 6 provides qualitative comparisons of object quantification performance across different styles. The red bounding boxes indicate failure cases where SDXL, IoCo, and QUOTA struggle to maintain accurate object counts, particularly in Painting and Cartoon styles. In contrast, *QuantiShift* consistently produces object counts that align more closely with the prompts, reducing errors across both Base and New domains. This demonstrates its ability to generalize to unseen shift conditions while maintaining high numerical fidelity.

## 6 CONCLUSION

We introduced *QuantiShift*, a shift-aware prompting framework that enhances object quantification in text-to-image diffusion models by addressing number shift, label shift, and covariate shift. Our approach combines shift-aware prompt optimization, consistency-guided adaptation, and hierarchical prompt optimization to improve numerical accuracy and generalization. To evaluate robustness under diverse shifts, we proposed *QSBench*, a benchmark specifically designed to assess object quantification performance. Experimental results demonstrated that *QuantiShift* significantly outperforms existing baselines.

**Ethics statement** This work studies object quantification in text-to-image diffusion models under distribution shifts. We do not collect or annotate any new human data. All components rely on *publicly available* resources under their respective licenses: a pretrained diffusion model (e.g., SDXL), a frozen counting estimator (CLIP-Count), and category/number/style specifications derived from public datasets (e.g., FSC-147 categories/styles) to define prompts and splits. Images used for optimization are *synthetically generated*; no personally identifiable information is processed. We employ GPT-based paraphrasing only to reword prompts without adding demographic or sensitive attributes. Our method targets numerical robustness (accurate counts) and does not infer or exploit identity-related attributes. Potential misuse includes generating misleading or restricted imagery (e.g., privacy-sensitive scenes or copyrighted content) with precise object counts. We discourage such practices and recommend deployments follow data-governance policies, model and dataset licenses, and content-safety filters (e.g., keyword blocking for sensitive categories). We report per-domain results to expose failure modes (e.g., style or category imbalance), and we constrain compute (single L20 GPU, §5) to keep the environmental footprint modest.

**Reproducibility statement** All datasets, pretrained models, and baselines are publicly accessible. We fully specify *shift-aware prompt optimization*, *consistency-guided prompting*, and *Hierarchical Prompt Optimization (HPO)* in §4.1, §4.2, and §4.3, including losses, update rules, and evaluation with paraphrased prompts. Experimental settings—splits (Base/New), styles, metrics (MAE/RMSE and harmonic means), hardware, and hyperparameters—are documented in §5. We provide ablations isolating tokens, consistency regularization, and HPO to facilitate replication. We will release source code, environment files with pinned versions, training/evaluation scripts, random seeds, and configuration files for all tables and figures. The release includes: (i) scripts to regenerate QSBench splits and prompts (categories, counts, and styles), (ii) precomputed paraphrases used for consistency experiments, (iii) reference prompt-token initializations and checkpoints, and (iv) instructions to reproduce every result from a fresh environment on a single-GPU machine.

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

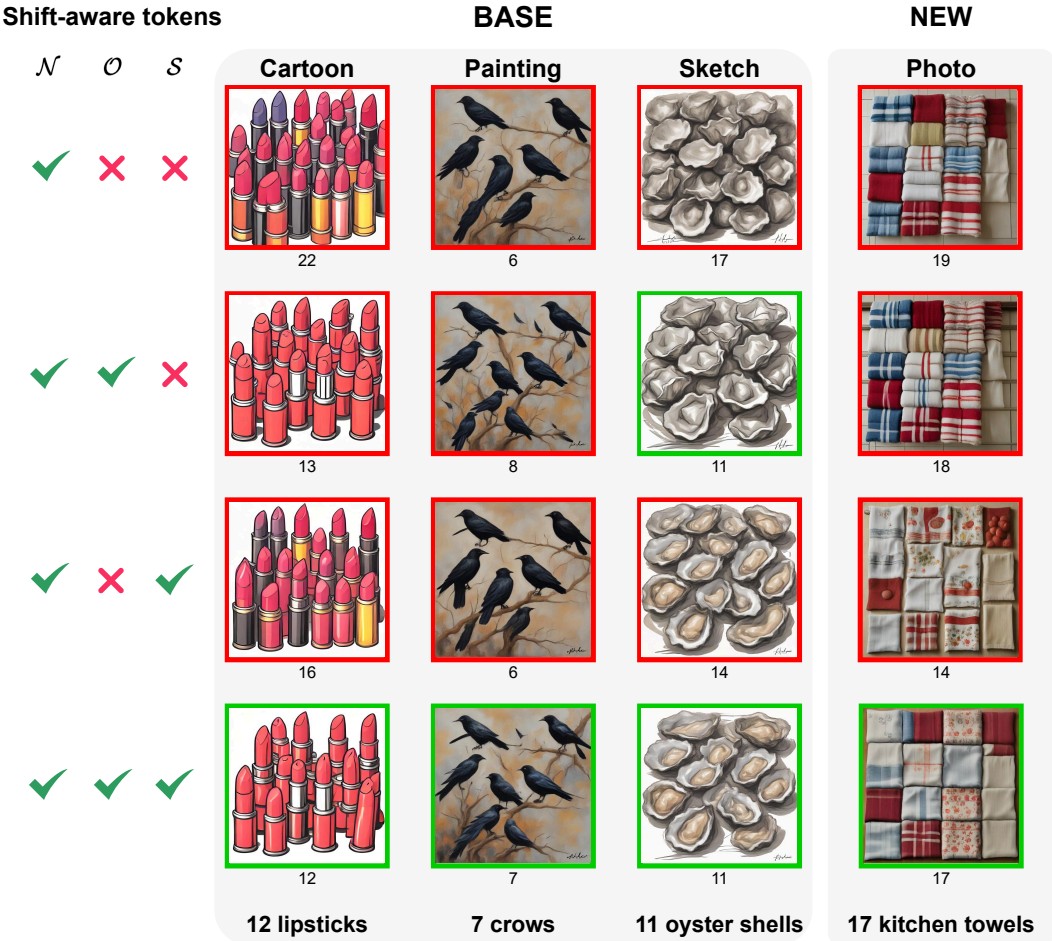

**Figure 7: Effect of different shift-aware tokens on object quantification.** We evaluate the impact of individual shift-aware tokens: $\mathcal{N}$ (number token), $\mathcal{O}$ (class token), and $\mathcal{S}$ (style token). The results show that using all tokens leads to the most accurate object counts across both base and new distributions, while individual tokens contribute to partial improvements depending on the shift type.

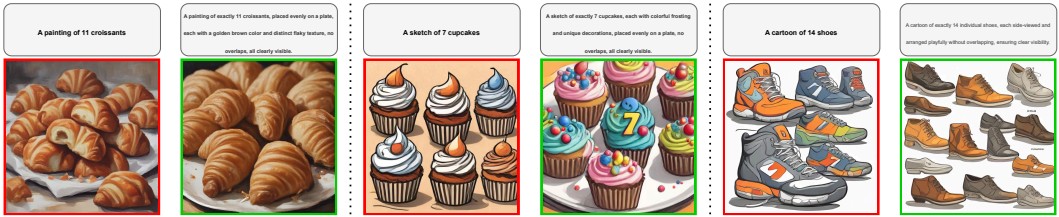

**Figure 8: Effect of detailed descriptions on numerical accuracy.** Simple prompts often produce incorrect object counts, while structured descriptions improve adherence to specified quantities by explicitly encoding numerical constraints and spatial arrangements.

## A   LLM USAGE STATEMENT

We used a large language model (ChatGPT) solely for grammar checking and language polishing of the manuscript text. It did not contribute to research ideation, method design, experiments, data analysis, or result generation; all technical content was authored and verified by the authors.

## B    DETAILED EXPERIMENTAL SETUP

***QSBench*: dataset splitting strategy.**   To rigorously evaluate the generalization ability of text-to-image diffusion models under number, label, and covariate shifts, we introduce *QSBench*, a benchmark explicitly designed with controlled splits. To create balanced splits, we first evaluated the generation difficulty of the SDXL model (Podell et al., 2024) across all 147 FSC-147 categories and numerical counts ranging from 1 to 25, totaling 3,675 tasks. For each task, we computed the Mean Absolute Error (MAE) between the generated and requested object counts. We then applied clustering based on these MAE scores along category and numerical dimensions separately, creating two subsets—*Base* and *New*—with similar cumulative MAE scores to ensure comparable difficulty levels. The resulting *Base* set contains 74 object categories and 12 numerical levels, while the *New* set comprises the remaining 73 categories and 13 numerical levels. Table 4 provides detailed category and quantity breakdowns for each subset.

**Covariate shift evaluation.**   To further evaluate generalization across visual styles, we consider four distinct styles: *Photo, Painting, Cartoon, Sketch*. Following a leave-one-out protocol, we perform four separate experiments, each time training on three styles (designated as *Base*) and evaluating on the remaining unseen style (*New*). Performance is assessed using MAE and RMSE metrics, and we report the harmonic mean to balance performance across seen and unseen styles. Finally, we average performance across all four scenarios to comprehensively measure the generalization capability of *QuantiShift*.

|  | Base Set | New Set |
|---|---|---|
| **Categories** | m&m pieces, geese, screws, sheep, cereals, coffee beans, horses, legos, red beans, pills, beads, bees, cashew nuts, grapes, mini blinds, chairs, matches, goats, birds, markers, cranes, fishes, naan bread, shoes, straws, cartridges, boxes, peppers, boats, caps, crows, stapler pins, cassettes, kidney beans, cans, prawn crackers, alcohol bottles, oranges, bottles, oyster shells, sticky notes, books, chopstick, elephants, lighters, finger foods, windows, keyboard keys, ice cream, spring rolls, calamari rings, skis, stairs, oysters, shallots, hot air balloons, jeans, croissants, people, instant noodles, cupcakes, rice bags, biscuits, watches, peaches, flowers, lipstick, meat skewers, baguette rolls, toilet paper rolls, potatoes, bread rolls, milk cartons, green peas | mosaic tiles, flamingos, candy pieces, buffaloes, stamps, goldfish snack, skateboard, cars, polka dot tiles, roof tiles, nuts, pigeons, pearls, ants, chewing gum pieces, crayons, seagulls, bricks, cows, polka dots, sea shells, zebras, swans, pencils, pens, watermelon, bottle caps, bananas, marbles, camels, gemstones, candles, supermarket shelf, cotton balls, tree logs, potato chips, coins, sunglasses, balls, cement bags, crab cakes, sauce bottles, clams, jade stones, tomatoes, nails, macarons, flower pots, deers, bullets, sausages, kiwis, fresh cut, spoon, penguins, cups, carrom board pieces, onion rings, go game, birthday candles, donuts tray, nail polish, strawberries, plates, chicken wings, bowls, kitchen towels, buns, shirts, cupcake tray, comic books, apples, eggs |
| **Numbers** | 1, 2, 7, 11, 12, 14, 15, 16, 20, 22, 24, 25 | 3, 4, 5, 6, 8, 9, 10, 13, 17, 18, 19, 21, 23 |

**Table 4: QSBench category and quantity splits for Base and New sets.** The Base set and New set contain non-overlapping categories and quantities while maintaining comparable cumulative MAE scores.

### B.1    DESCRIPTION GENERATION

To enhance the alignment between text prompts and generated images, we optimize the original prompt structure $\langle A, S, \text{of } N \text{ CLASS}, O \rangle$ using GPT-4o. The goal is to ensure that the generated images accurately reflect the specified quantity (N) and category (O) while maintaining clarity, correctness, and consistent arrangement. We employ the following instruction for GPT-4o (OpenAI, 2023) to generate structured, optimized descriptions:

**Prompt to GPT-4o:**

> *I want to generate optimized Prompts starting from a template: 'A S of N O'. The objective is to ensure the generated images accurately reflect the specified quantity (N) and category (O), with a focus on clarity, correctness, and consistent arrangement. Each O should be well-described in terms of appearance and arrangement, avoiding overlaps, semantic ambiguity, or inclusion of unrelated*

*objects or complex backgrounds. The range for N is from 1 to 25, and S can be one of four options: 'photo', 'painting', 'sketch', or 'cartoon'. For each O, create an optimized Prompt Bank of 10 entries that describe the objects clearly and guide their spatial arrangement to support accurate generation.*

For each category, we construct a description template bank consisting of 10 entries. During generation, a description is randomly sampled from the template bank, with placeholders replaced to match the required style (S) and quantity (N).

**Example category-specific description templates:**

- **Flowers:**
  - A S of exactly N flowers, each with a different color, arranged symmetrically in a vase, no overlaps, all clearly visible.
  - A S showing N flowers, each with distinct colors, neatly arranged in a row on a flat surface, no overlaps, all visible.
  - A S of N flowers, each with unique colors and types, placed evenly on a flat surface, no overlaps, all visible.
  - A S showing exactly N flowers, each with vibrant petals and different colors, arranged symmetrically, no overlaps, all visible.,
  - A S of N flowers, placed symmetrically on a flat surface, each with a unique color, no overlaps, all visible.
- **Cupcakes:**
  - A S of exactly N cupcakes, each with colorful frosting and unique decorations, placed evenly on a plate, no overlaps, all clearly visible.
  - A S showing N cupcakes with bright, colorful frosting and sprinkles, arranged symmetrically on a flat surface, no overlaps, all visible.
  - A S of N cupcakes, each with a different frosting design and color, placed neatly in a row, no overlaps, all visible.
  - A S showing exactly N cupcakes with distinct frosting designs and colors, arranged symmetrically on a plate, no overlaps, all visible.
  - A S of N cupcakes, placed evenly on a plate, each with unique frosting designs and decorations, no overlaps, all visible.

By using GPT-4o to refine the prompts, QSBench ensures that the generated images align more accurately with their intended attributes, improving consistency in object count, style, and arrangement.

### B.2    TRAINING DETAILS

Each experiment consists of 5 epochs, with each epoch comprising 2,664 iterations, determined by the total number of category-quantity-style combinations in the Base set ($74 \times 12 \times 3$). This results in a total of 13,320 iterations per experiment.

To prevent overfitting and improve generalization, we employ an independent random sampling strategy for each iteration. Rather than iterating over all possible combinations in a fixed order, our approach ensures that each training instance is drawn stochastically, introducing variability in the data and making the model more robust to unseen samples. It is important to note that this random sampling strategy is applied only during the training phase. In the evaluation phase, both Base and New sets are tested exhaustively, ensuring full coverage of all category-quantity-style combinations.

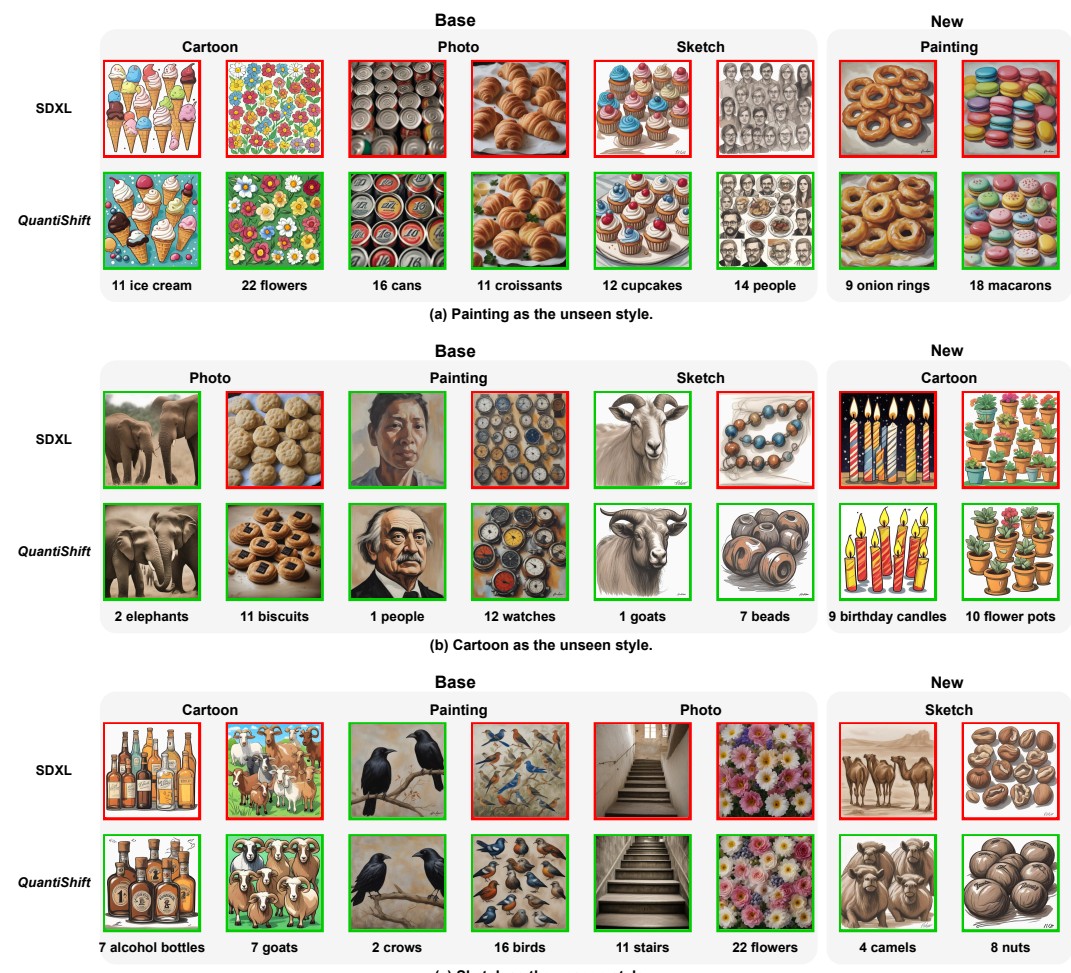

Figure 9: **Qualitative comparison of object quantification performance across different visual styles.** We compare the generated images from SDXL and *QuantiShift* when generalizing to unseen visual styles: Painting, Cartoon, and Sketch. The red bounding boxes highlight failure cases where SDXL generates incorrect object counts. Our approach, *QuantiShift*, demonstrates improved generalization and more accurate quantity preservation across diverse styles.

Specifically, the Base set evaluation consists of $74 \times 12 \times 3 = 2,664$ test cases, while the New set evaluation includes $73 \times 13 = 949$ test cases. This comprehensive testing approach ensures that model performance is assessed fairly across all possible conditions.

Each training iteration takes approximately 0.1 minutes, leading to an estimated total training duration of around 22.2 hours per experiment. The exact time may vary depending on the styles present in the Base set. Compared to IoCo (Zafar et al., 2024), which trains separate models for each category-quantity-style combination—requiring approximately 88.8 hours to complete similar training tasks—our method is significantly more efficient while maintaining model adaptability.

We use the AdamW optimizer with a learning rate of 0.01 to ensure stable convergence throughout training, balancing effective learning dynamics and preventing excessive parameter updates.

## C    MORE RESULTS

**Qualitative Analysis** To further demonstrate the effectiveness of *QuantiShift* in handling object quantification across different visual styles, we conduct additional qualitative comparisons where Painting, Cartoon, and Sketch are treated as unseen styles. These comparisons focus on evaluating the

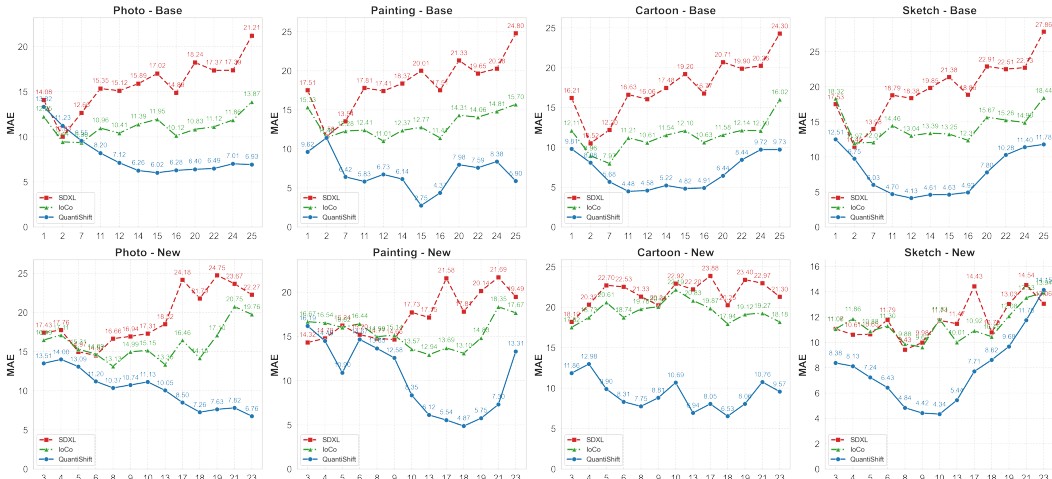

**Figure 10: Analysis of object count difficulty across domains.** We visualize the relationship between object quantity and MAE for the Base and New subsets under four distinct styles (*Photo, Painting, Cartoon, Sketch*). The results show that quantifying objects accurately becomes increasingly challenging for both SDXL and our method as the object quantity increases. However, *QuantiShift* consistently maintains significantly lower MAE across all numerical ranges, especially in challenging mid-to-high quantity scenarios, demonstrating robust handling of numerical variations compared to SDXL.

object quantity accuracy and overall image quality of *QuantiShift* against SDXL. Figure 9 presents visual results across these unseen styles. As highlighted by the red bounding boxes, SDXL frequently fails to maintain the correct object count, particularly when the required quantity is large. In contrast, *QuantiShift* consistently produces images with more accurate object quantities while preserving visual quality and diversity. Interestingly, we observe that SDXL can accurately generate images with smaller object quantities. For example, in Figure 9 (b), cases such as "A photo of 2 elephants," "A painting of 1 person," and "A sketch of 1 goat" exhibit correct object counts. However, as the specified number increases, the deviation becomes more pronounced, further emphasizing the advantage of *QuantiShift* in handling complex quantity control. In the following section, we provide a quantitative analysis using line plots to measure the MAE across different object quantities in each experiment.

**Analysis of object quantification difficulty across quantities.** Figure 10 analyzes how the difficulty of object quantification varies with the specified quantity of objects. We observe a clear trend: as the number of objects increases, our method, the SDXL baseline, and IoCo (Zafar et al., 2024) all face higher quantification errors, reflecting the intrinsic difficulty of counting more objects in complex visual scenes. Despite this, *QuantiShift* consistently achieves lower MAE values compared to SDXL and IoCo, particularly for mid-range (8-15) and high-range (16-25) quantities. This highlights the effectiveness of our shift-aware and consistency-guided prompting strategies, which significantly improve numerical accuracy by mitigating ambiguities associated with higher counts. These findings confirm that our method enhances robustness and reliability in object quantification, particularly in scenarios involving higher object quantities that typically pose substantial challenges.

