# OpenReview forum: "QuantiShift: Any-Shift Object Quantification by Text-to-Image Diffusion Models"
_ICLR.cc/2026/Conference — ICLR 2026 Conference Withdrawn Submission_

### Official Review · Reviewer_Uhd2 · 2025-10-27

**Soundness:** 3
**Presentation:** 3
**Contribution:** 3
**Rating:** 4
**Confidence:** 4

**Summary:**

The paper introduces QuantiShift, a framework designed to solve text-to-image models' inability to generate specific number of objects, especially when the prompt includes new object types, counts, or styles the model hasn't seen before. Quantishift works by optimizing the text prompt with its three components:
- Shift-aware prompt optimization: Quantishift creates a structured prompt with separate, learnable tokens that act as independent control knobs for the object's count, category, and style.
- Consistency-guided prompting: To make the model to more robust to prompt variety, they utilized an external LLM to rewrite the prompts. It then trains the model to create the nearly identical images for both the original and the paraphrased prompt.
- Hierarchical prompt optimization: QuantiShigt first learns to handle each shift type in isolation, and subsequently calibrates the prompt to perform well on many shift scenario.

The paper also introduces new evaluation benchmark called QSBench to test object quantification under diverse shifts.

**Strengths:**

- No model retraining required. Learning prompt tokens (number/class/style) is a lightweight way to condition a pre-trained generator for counting, avoiding the substantial compute of fine-tuning and improving practicality/scalability.
- Object miscounting is a persistent failure mode of diffusion models; evaluating robustness under distribution shifts (number, label, style) is well-motivated.
- The method explicitly targets multiple shift sources rather than only domain/style, aiming for broader generalization.
- The two-stage (“local then global”) refinement is a sensible meta-learning-style strategy to reduce overfitting to individual shift conditions.
- QSBench varies factors systematically and shows clear gains in count accuracy across shift conditions, suggesting the approach generalizes beyond a single domain.

**Weaknesses:**

- Several ingredients closely follow prior lines of work: learnable/pseudo-token prompting (cf. textual inversion and prompt tuning), dual-loop/meta-optimization (eg. QUOTA-style formulations), and paraphrase-based consistency/contrastive alignment. The main advance is their combination for "any-shift" counting rather than a fundamentally new algorithmic idea.
- Results emphasize individual shift types. It remains unclear how the method fares under concurrent shifts (e.g., unseen class + extreme count + unseen style). The paper also acknowledges failures when explicit counts collide with implicit numeric attributes (e.g., "4 90-year-olds"), suggesting remaining entanglements.
- The optimization process is designed to minimize a counting loss provided by a frozen CLIP-Count model. This means the framework isn't learning to generate images with the true number of objects but it is learning to generate images that deceive the CLIP-count model into reporting the correct number.
- Beyond SDXL, IoCo, and QUOTA, recent counting/control approaches (e.g., Binyamin et al., Make it Count) or attention-constraint methods could be discussed or compared where applicable. Without this, the comparative picture feels narrower than the claim.
- The paper focuses on count errors (MAE/RMSE) but does not quantify effects on fidelity/diversity. Are improvements achieved at the cost of artifacts or mode collapse? Qualitative figures alone are insufficient to rule this out.
- Critical hyperparameters and training choices are under-specified (e.g., paraphrase generation protocol/filters, consistency loss formulation and weight, inner/outer-loop schedules/steps, token initialization). Given three coupled components + a new benchmark, clearer algorithms/pseudocode or code release is important.
- It is not explicit whether QSBench design and hyperparameter selection were isolated from final evaluation (to avoid latent leakage/overfitting to the benchmark construction itself).
- A deeper examination of where QuantiShift breaks (very large counts, heavy occlusion/overlap, fine-grained categories, strong style gaps) would make the robustness claims more convincing.
- The related work section should be extended to include more recent work on "Prompt optimization for image consistency". The most recent work cited in this section is from 2023.

**Questions:**

- How are number/class/style tokens instantiated and optimized (e.g., learned embeddings prepended/appended to the text sequence)? Are they global tokens reused across prompts or category-specific?
- Do you evaluate cases that combine number + label + style shifts simultaneously? If yes, please report; if not, what failure modes emerge when shifts are compounded?
- Will QSBench (prompts/splits) be released? Were hyperparameters tuned on a separate validation split? Please clarify hold-out practices to avoid benchmark overfitting.
- You chose SDXL, IoCo, and QUOTA as baselines, which makes sense. However, there are other related works (e.g., CountGen, CONFORM, etc.). Did you consider any of these in either the related work or experimentation?

---

### Official Review · Reviewer_q1R2 · 2025-10-28

**Soundness:** 2
**Presentation:** 2
**Contribution:** 2
**Rating:** 4
**Confidence:** 3

**Summary:**

The paper proposes to improve the counting, object label grounding and visual style aligning ability of diffusion models by prompt optimization. Specifically, it proposes to use three set of special tokens: number token (N), class token (O) and style token (S) to learn the concept of specific number, class type or visual style. During inference, the special token can be reused to augment the sampling quality.  The paper further develops hierarchical prompt optimization to jointly optimize for the tokens. To evaluate the effectiveness of the proposed method, the paper further introduces a new benchmark designed to assess the robustness under different prompt distribution shifts.

**Strengths:**

+ The paper proposes an effective solution for prompt optimization and extends the previous paradigm to control across more fine-grained specific visual domains.
+ The paper propose hierarchical optimization techniques to further improve the effectiveness of the proposed method.
+ The paper conducted thorough experiments for ablating the proposed hierarchical optimization framework and jointly using three types of special tokens.
+ The paper is generally clearly written and easy to follow.

**Weaknesses:**

- The idea of prompt optimization or introducing does not seem entirely novel. As mentioned also in the paper (sec. 2), for example, textual inversion (Gal et al., 2023) has already introduced similar techniques, which shares the core idea with this paper of using special tokens. The novelty of the paper therefore seems largely resides on the hierarchical optimization technique, which appears to be limited.
- The evaluation metrics seem a bit uncommon to me. Can the author provide more justification why MAE and RMSE are used for measuring the performance, instead of using accuracies for object binding and counting?
- The experiment results are heavily concentrated on the SD-XL model. How does the proposed technique work with more advanced architectures and models such as sd3.5-M/L or flux models? Direct results based on these models could further improve the positioning of this manuscript and help with assessment.
- Admittely, I am not very familiar with the current progress of this particular sub-domain. I would therefore also like to hear other colleague reviewers' opinions.

**Questions:**

Please refer to the weaknesses section for detailed questions. Thanks.

---

### Official Review · Reviewer_7Nxq · 2025-10-31

**Soundness:** 2
**Presentation:** 2
**Contribution:** 2
**Rating:** 4
**Confidence:** 4

**Summary:**

This paper tackles the problem of object quantification problem in text-to-image diffusion models. Speicifcally, they propose QuantiShift, a shift-aware framework that performs image generation with accurate object quantification.

**Strengths:**

1. The paper tackles an interesting problem of enabling pretrained diffusion models to produce images with accurate object quantifiactions.
2. The performance of the proposed method seems good.

**Weaknesses:**

1. The paper’s motivation is unclear and not well developed. While the authors position “object quantification under distribution shifts” as an emerging problem, the paper does not convincingly explain why this task matters, who would need it, or what real-world scenarios it enables.
2. The training and testing pipeline is not clearly described, e.g., do they need to train new prompt each time a new style/object is introduced? Without a transparent description of how tokens are trained, reused, or adapted, it is impossible to determine whether QuantiShift truly handles “any-shift” generalization or merely performs domain-specific fine-tuning.
3. Both training supervision and performance measurement rely on CLIP-Count, an external model that itself can make mistakes, especially for stylized or non-photorealistic images. The paper does not validate whether CLIP-Count correlates with human judgments, so improvements may reflect artifacts of the proxy rather than genuine gains in quantification.
4. The paper does not clearly introduce the proposed QSBench, except that it is built upon QUANT-bench and extends it with FSC147. The size of the training/testing set is not clearly described. Also, FSC147 contains mostly natural style images.
5. The evaluation metrics only include object quantification-related ones. The quality of the generated images are not evaluated.

**Questions:**

Please refer to the weakness part.

---

### Official Review · Reviewer_3wks · 2025-11-01

**Soundness:** 2
**Presentation:** 2
**Contribution:** 2
**Rating:** 2
**Confidence:** 3

**Summary:**

The paper “QuantiShift: Any-Shift Object Quantification by Text-to-Image Diffusion Models” addresses the challenge of accurately generating and counting objects under distribution shifts in text-to-image diffusion models. Existing models often fail to preserve object quantities when the number, category, or visual domain differs from training data. The authors propose QuantiShift, a shift-aware prompting framework that adapts text prompts rather than retraining models. It introduces (1) shift-aware prompt optimization to handle number, label, and domain shifts, (2) consistency-guided any-shift prompting to ensure alignment between text and generated images, and (3) a hierarchical optimization strategy for cross-shift generalization. Experiments on diverse benchmarks demonstrate that QuantiShift significantly improves counting accuracy and robustness compared to existing methods, without requiring model fine-tuning. The work contributes a new perspective on prompt-level adaptation for robust visual quantification and introduces an “any-shift” benchmark to evaluate model reliability under distribution shifts.

**Strengths:**

1. The paper introduces a novel shift-aware prompting framework that tackles the underexplored problem of object quantification under distribution shifts in text-to-image diffusion models.

2. The paper clearly articulates its motivation, provides a new dataset, and experimental evaluation on diverse benchmarks.

**Weaknesses:**

1. While the paper presents a well-organized prompting framework, its contribution appears incremental over the prior AnyShift work, primarily extending the concept from the community of CLIP  into diffusion generation. The methodological changes build naturally on existing shift-aware ideas rather than introducing fundamentally new mechanisms.

2. Moreover, the problem formulation is narrowed to counting scenarios, which limits the general applicability of the approach to broader generative reasoning or compositional understanding tasks.

3. The paper lacks direct comparison with the AnyShift baseline (with minimal adaptation to diffusion generation), leaving unclear whether the gains stem from the proposed optimization or differences in setup.

4. The effectiveness of the bi-level optimization procedure is also insufficiently justified—no ablation over key hyperparameters (e.g., weighting factor $\alpha$) or components (constraint vs. objective) is provided.

5. Furthermore, the method lacks theoretical grounding or convergence analysis to explain why the hierarchical prompt optimization should improve robustness under distribution shifts.

**Questions:**

1. Please refer to the Weakness for my questions.

2. There are existing counting tasks in benchmarks such as CompBench [1] and GenEval [2]. A fair comparison with other counting enhancement methods—or more general alignment enhancement approaches—would significantly improve the paper’s experimental rigor and credibility. Including such baselines would help demonstrate that the proposed framework provides consistent advantages beyond the specific setting introduced by the authors.

[1] CompBench: Benchmarking Complex Instruction-guided Image Editing

[2] GenEval: An Object-Focused Framework for Evaluating Text-to-Image Alignment

---

### Note · Authors · 2025-11-12

I have read and agree with the venue's withdrawal policy on behalf of myself and my co-authors.